# The Role of Macronutrients and Gut Microbiota in Neuroinflammation Post-Traumatic Brain Injury: A Narrative Review

**DOI:** 10.3390/nu16244359

**Published:** 2024-12-18

**Authors:** Antonella Cotoia, Ioannis Alexandros Charitos, Alberto Corriero, Stefania Tamburrano, Gilda Cinnella

**Affiliations:** 1Department of Intensive Care, University Hospital of Foggia, 71121 Foggia, Italy; stef.tambu@gmail.com (S.T.); gilda.cinnella@unifg.it (G.C.); 2Istituti Clinici Scientifici Maugeri IRCCS, Pneumology and Respiratory Rehabilitation Unit, “Istitute” of Bari, 70124 Bari, Italy; alexanestesia@hotmail.com; 3Doctoral School on Applied Neurosciences, Dipartimento di Biomedicina Traslazionale e Neuroscienze (DiBraiN), University of Bari “Aldo Moro”, 70121 Bari, Italy; 4Department of Interdisciplinary Medicine-ICU Section, University of Bari “Aldo Moro”, Piazza Giulio Cesare 11, 70124 Bari, Italy; alberto.corriero@gmail.com

**Keywords:** neuroinflammation, traumatic brain injury, gut microbiota, polyunsaturated fatty acids, probiotics

## Abstract

Traumatic brain injury (TBI) represents a multifaceted pathological condition resulting from external forces that disrupt neuronal integrity and function. This narrative review explores the intricate relationship between dietary macronutrients, gut microbiota (GM), and neuroinflammation in the TBI. We delineate the dual aspects of TBI: the immediate mechanical damage (primary injury) and the subsequent biological processes (secondary injury) that exacerbate neuronal damage. Dysregulation of the gut–brain axis emerges as a critical factor in the neuroinflammatory response, emphasizing the role of the GM in mediating immune responses. Recent evidence indicates that specific macronutrients, including lipids, proteins, and probiotics, can influence microbiota composition and in turn modulate neuroinflammation. Moreover, specialized dietary interventions may promote resilience against secondary insults and support neurological recovery post-TBI. This review aims to synthesize the current preclinical and clinical evidence on the potential of dietary strategies in mitigating neuroinflammatory pathways, suggesting that targeted nutrition and gut health optimization could serve as promising therapeutic modalities in TBI management.

## 1. Introduction

Traumatic brain injury (TBI) refers to an alteration in brain function caused primarily by external forces. The primary causes of TBI are external forces that directly impact the skull’s intracranial surface, potentially harming the soft brain tissue in contact with it. Traumatic axonal injury is a specific type of TBI that involves widespread damage to the brain’s axonal pathways, often due to high-velocity translational or rotational forces during rapid motion changes, typically due to a motor vehicle accident or fall [1].

Conversely, non-impact events, like blast waves or rapid changes in speed, can also lead to TBI without requiring direct contact with the head. Acquired injuries from TBI can lead to changes in various structural components of the brain, resulting in either temporary or lasting cognitive and functional impairments. Factors like the severity and location of the injury, frequency of occurrence, and individual characteristics such as sex, age, genetics, and existing health conditions influence the neurological effect of TBI. Consequently, the complexity of TBI outcomes stems from the immediate mechanical damage caused by the injury (primary injury) and the subsequent delayed molecular responses that follow (secondary injury). Unfortunately, TBI remains one of the leading causes of injury-related deaths, disabilities, and mental health issues, posing a significant public health challenge globally [2,3,4,5,6,7].

TBI is a complex disease that impacts multiple organs, such as the lungs, gastrointestinal system (GI), liver, and kidneys. The gut–brain axis (GBA) facilitates the bidirectional pathway between the brain and GI system through a sophisticated network involving neurons, hormones, and immune responses. This interaction primarily occurs via four key pathways: the immune system, autonomic nervous system (ANS), enteric nervous system (ENS), neuroendocrine system, and microbiota.

Recent evidence suggests that the lung microbiota may also influence immune reactivity of the brain and systemic inflammation, which can exacerbate neuroinflammatory responses following TBI, highlighting the existence of a lung–brain axis [8,9].

Additionally, specific nutrients can help reshape the GM, potentially mitigating the effects of second-hit injuries and supporting the brain’s resilience after TBI.

By integrating the complex relationships between dietary intake, microbiota composition, and the resulting neuroinflammatory responses, we aim to provide preclinical and clinical evidence that these dietary interventions can effectively promote neurological recovery.

## 2. Methods

For this narrative review, we conducted a comprehensive search across several databases, including PubMed, Web of Science, Google Scholar, SpringerLink, and ScienceDirect. The search utilized specific keywords such as “neuroinflammation” and “traumatic brain injury”, which were combined with related terms like “microbiota”, “macronutrients”, “protein”, “amino acid”, “lipids”, “short-chain fatty acids”, “probiotics.” The final search was carried out in September 2024, focusing exclusively on literature published in English.

References were selected based on their relevance to the review topic. The process began with a review of the titles of key references, followed by a detailed examination of their content. Articles that did not pertain to the review or that involved duplicated studies were excluded. Additionally, the abstracts of the remaining articles were assessed to confirm that they aligned with the inclusion criteria set for this review. As this is a narrative review, detailed documentation of the literature searches across specific platforms is not required.

## 3. Pathophysiology of Traumatic Brain Injury

Primary injury involves the displacement and mechanical damage of brain tissue, manifesting as contusions, vascular damage, hemorrhages, changes in cerebral blood flow, alterations in blood–brain barrier (BBB) permeability, and metabolic disturbances [10].

Following this initial injury, secondary brain injury occurs, marked by a series of complex biochemical processes that begin within minutes and can persist for days, months, or even years [1]. This secondary response exacerbates mitochondrial dysfunction, neuroinflammation, neurodegeneration, and neurological impairments. Moreover, the pathophysiology of TBI involves both immediate necrotic and subsequent apoptotic death of neurons [7].

Key processes in secondary injury include the ongoing imbalance of intracellular ions (such as Ca^2+^, Na^+^, and K^+^) and the excess release of neurotransmitters like glutamate leading to excitotoxicity. Excitotoxicity has long been recognized as a mechanism of secondary brain injury that contributes to neuronal death after trauma. This form of cell death is closely associated with the modulation and dysregulation of calcium ion influx, with calpains playing a significant role in this process [1].

The activation of glutamate receptors and voltage-gated calcium (Ca^2+^) channels leads to an influx of Ca^2+^, which facilitates the release of fatty acids (FAs) from membrane phospholipids. This release can be detected in cerebrospinal fluid (CSF) for up to one week following a TBI [11].

Moreover, proteins from damaged cells can escape into the cytoplasm and potentially affect cellular functions that activate both central and systemic immune responses. The compromised BBB facilitates the entry of peripheral immune cells, including leukocytes, into the brain’s parenchyma. These immune cells release chemokines and cytokines, which stimulate and mobilize resident glial cells like microglia and astrocytes, further promoting the infiltration of additional immune cells [10].

Polyunsaturated fatty acids (PUFAs) derived from these membrane lipids can be preferentially converted into bioactive lipids that modulate inflammation. Notable examples include arachidonic acid (AA), docosahexaenoic acid, and eicosapentaenoic acid. AA, an omega-6 PUFA, serves as a critical precursor for pro-inflammatory eicosanoids such as prostaglandin E2 and leukotriene B4 [12,13].

The secondary injury mechanisms after TBI trigger responses that extend beyond the injured brain tissue, affecting peripheral organs, including the GI system. The GM is crucial for immune cell populations and microglial activity regulation, and dysbiosis can lead to immune system dysfunction and behavioral issues. Nevertheless, alterations in the GM resulting from acute TBI have not been extensively studied [14]. In a study examining the impact of TBI on GM composition using a controlled cortical impact model in male C57BL/6J mice, significant changes in both genus- and species-level microbiota were observed 24 h post-injury compared to baseline. Two significant changes were identified at the genus level: an increase in *Marvinbryantia* and alterations within the *Clostridiales*. At the species level, three species exhibited significant decreases: *Lactobacillus gasseri*, *Ruminococcus flavefaciens*, and *Eubacterium ventriosum*. Conversely, two species showed significant increases: *Eubacterium sulci* and *Marvinbryantia formatexigens*. These findings highlight an acute dysbiosis in the GM following TBI, characterized by specific shifts in microbial communities, which may play a role in the broader physiological effects of brain injury (Figure 1) [14,15].

A recent study investigated changes in GM in fecal samples of patients with intracerebral hemorrhage (ICH) at three time points, within 24 h of admission and 3 and 7 days post-surgery, comparing them to healthy controls using 16S rRNA sequencing. Increased abundances of *Enterococcaceae*, *Clostridiales incertae sedis XI*, and *Peptoniphilaceae* were observed in ICH patients, while *Bacteroidaceae*, *Ruminococcaceae*, *Lachnospiraceae*, and *Veillonellaceae* were reduced. The relative abundance of *Enterococcus* increased after surgery, whereas *Bacteroides* decreased. A higher abundance of *Enterococcus* before surgery was associated with worse neurological outcomes, while *Lachnospiraceae* was identified as having a potential protective effect. Overall, alterations in GM diversity were linked to the prognosis of ICH patients, suggesting a relationship between gut health and neurological recovery [16]. The GBA entails bidirectional pathways where TBI-induced neuroinflammation and neurodegeneration impact gut function. This leads to dysautonomia and systemic inflammation, contributing to GI issues, including dysmotility and increased mucosal permeability. Furthermore, these changes are influenced by dysbiosis and immune cell activity. Microbial products and immune mediators can affect brain–gut interactions, while secondary enteric inflammatory challenges exacerbate systemic inflammation, worsening TBI-related neurological and behavioral deficits. Thus, maintaining brain–gut communication is crucial and presents potential strategies in TBI treatment. Neuroinflammation has been called a “chronic response to an acute injury” [17], and the hypothesis that attenuation of acute neuroinflammation after TBI can reduce short- and long-term disability by limiting secondary injury and promoting CNS recovery is attractive and is the basis of several therapeutic approaches described in experimental TBI models and clinical research.

## 4. Gut Microbiota’s Role in Central Nervous System (CNS)

The GBA is the connection and two-way communication between the ENS, also known as the intrinsic nervous system, and the CNS via the vagus nerve (also called the pneumogastric nerve). Afferent nerve fibers of the vagus nerve can be activated either via enteroendocrine cells releasing hormones or by the GM, conveying information to the brainstem [18].

The vagus nerve, thought to fulfill anti-inflammatory actions, carries information back to the gut through its efferent fibers. It can indirectly modify the GM composition by influencing mucus secretion, intestinal barrier permeability, and immunological responses. These actions are primarily carried out by releasing of neurotransmitters such as acetylcholine and catecholamines [19].

Acetylcholine released from the vagus nerve appears to reduce the secretion of pro-inflammatory cytokines such as TNF-α and IL-6. The importance of the vagus in GBA axis communication is demonstrated by studies in vagotomy experimental animals. Indeed, the anxiolytic effect of *Bifidobacterium longum* requires the integrity of the vagus nerve and was not mediated in its absence in an experimental model of chronic colitis [20,21].

Similarly, the administration of *Lacticaseibacillus rhamnosus* (old name *Lactobacillus rhamnosus*) to experimental animals reduces anxious and depressive behavior by modulating the expression of the γ-aminobutyrate (GABA) receptor in the brain. These effects were not found in animals that underwent vagotomy [22].

Thus, neurotransmitters and other neuroregulatory hormones primarily mediate the neuroendocrine signaling pathway. This pathway is essential for metabolic homeostasis and influences emotions, mood, and, in general, higher cognitive and organic CNS functions. It is a complex system involving the CNS, the ANS, the brain, the spinal cord, and the hypothalamic–pituitary–adrenal (HPA) axis [23].

The HPA axis is another neuroendocrine signaling pathway that connects the brain and the GM and regulates the stress response. Several studies show that the GM is essential for the development and maturation of this axis, as sterile experimental animals exhibit a pathologically increased stress response [24].

Conversely, the activation of the HPA axis and the subsequent secretion of glucocorticoids can modify the composition of the GM, interfering with mucus production, gastrointestinal motility, and intestinal epithelial permeability. Glucocorticoids can also function as immune response regulators, having anti-inflammatory and pro-inflammatory effects on peripheral immune cells and CNS microglial cells [24].

Thus, the GBA indicates a bidirectional relationship and interdependence between the CNS and ENS.

It is important to understand this interaction and how a healthy GM (eubiosis) can affect the body and nervous system, and vice versa. Through several pathway mechanisms and a wide array of substances and neurotransmitters, the relationship between the two complex systems is constantly changing, which can have both positive and negative effects [23]. Inter-communication between the CNS and bacteria in the GM relies on the presence of neurotransmitter receptors in the bacteria. Several studies have reported that binding sites for host-produced enteric neurotransmitters exist in bacteria and can affect the function of microbiota, contributing to increased susceptibility to inflammatory and infectious stimuli. High affinity has been reported for the GABA system in *Pseudomonas fluorescens*, showing binding properties similar to those of a brain receptor [25] (Figure 2).

*Escherichia coli* O157:H7 possesses a receptor for host-derived epinephrine/norepinephrine that can be specifically blocked by adrenergic antagonists [26].

Acute stress can increase colonic permeability, leading to overproduction of interferon-γ (INF-γ). An example is the secretion of norepinephrine during surgery that causes the expression of *Pseudomonas aeruginosa*, which can lead to intestinal sepsis. In addition, norepinephrine can stimulate the proliferation of enteric pathogens and increase the infectious properties of *Campylobacter jejuni* [25,27,28].

The brain can influence the composition and function of the GM by altering intestinal permeability, allowing bacterial antigens to penetrate the epithelium and stimulate an immune response in the mucosa. Acute stress increased colonic paracellular permeability associated with overproduction of INF-γ and decreased mRNA expression of the tight junction proteins ZO-2 and occludin [29,30].

The brain can also modulate immune response via the ANS. The sympathetic branch modulates the quantity and the activity of mast cells with consequent imbalance in tryptase and histamine release in stress-related muscle dysfunction [31].

Other mast cells products, such as the corticotropin-releasing factor (CRF), can increase epithelial permeability to bacteria, thereby facilitating their access to immune cells. Furthermore, corticotropin-releasing hormone receptors play a role in disrupting the colonic barrier following mild stress during periods of neonatal maternal separation in adult rats, contributing to depressive symptoms and increased susceptibility to colitis [32].

Bilateral olfactory bulbosection induced depression-like behavior associated with increased central CRF expression and serotonin levels, associated with changes in colonic motility and gut microbial profile in mice [33].

Dysbiosis (unhealthy changes in the quality and quantity of microorganisms in the GM) is highly associated with mood disorders and linked to GBA disruptions. It has been noted that dysbiosis occurs after brain injury, characterized by an increase in *Pseudomonadota* (formerly known as *Proteobacteria*), particularly within the *Enterococcaceae* family (such as *Enterococcus* spp.), as well as *Lachnoclostridium* and *Parabacteroides* spp. Additionally, there is a decrease in *Bifidobacterium*, *Faecalibacterium*, and *Akkermansia muciniphila* [34,35].

The GBA imbalance causes changes in intestinal motility and secretion, induces visceral hypersensitivity, and leads to cellular alterations of the enteroendocrine and immune systems. Finally, it is important to note that gut changes associated with stress facilitate the proliferation of virulent bacteria. Norepinephrine released during surgery induces the expression of *Pseudomonas aeruginosa*, which can lead to intestinal sepsis [36].

In addition, norepinephrine can also stimulate the proliferation of various strains of enteric pathogens and increase the virulence properties of *Campylobacter jejuni*. It can also favor the overgrowth of both non-pathogenic *E. coli* isolates and pathogenic *E. coli* 0157: H7:3.

Finally, *Clostridium butyricum* and *Lactobacillus acidophilus* have been shown to have neuroprotective effects in TBI models in mice and can help mitigate neuroinflammation and impaired neurogenesis post-TBI [37,38].

## 5. The Role of the Gut Microbiota’s Metabolome in Neurotransmission

Serum short-chain fatty acid (SCFA) metabolites/conjugates, serum amino acids (SAAs), and serum bile acid (SBA) metabolites/derivatives are the three major groups of the most studied GM-related metabolites in health and disease [23].

The GM plays an important role in the metabolism of amino acids, the products of which can affect the synthesis of certain neurotransmitters. Over 90% of total body serotonin is produced in the gut by enterochromaffin cells, a subset of enteroendocrine cells, via the enzyme tryptophan hydroxylase. Serotonin (5-hydroxytryptamine, 5-HT) exerts its effects through receptors on neighboring epithelial cells, immune cells and neurons of the ENS, increasing peristalsis [39].

Some bacteria can regulate metabolites that favor serotonin biosynthesis in the colon, possibly through increasing the expression of tryptophan hydroxylase, while studies have shown that sterile experimental animals show lower circulating serotonin concentrations [40,41].

In addition, it is known that some bacteria can produce neurotransmitters de novo, e.g., *Bifidobacterium* strains and those from the *Lactobacillaceae* family can produce γ-aminobutyrate, the main inhibitory neurotransmitter, while other species can synthesize dopamine, norepinephrine, and acetylcholine [42].

Bacterial metabolites also have a role in GBA communication. SCFAs are metabolites produced by microbial fermentation of non-digestible polysaccharides, including dietary fibers and resistant starches in the distal gut. The most prevalent SCFAs are acetate, propionate, and butyrate, which collectively account for over 95% of total SCFAs [43]. Acetate production requires substrates described as acetogenic fibers (inulin, galactooligosaccharides, etc.) [44]. Most of the enteric bacteria, such as *Akkermansia muciniphila*, *Bacteroidota phyla* spp., or *Prevetolla* spp., are acetate producers [45,46].

Propionate is primarily produced by bacteria from the *Bacteroidota* and *Bacillota* phyla [45].

While most butyrate-producing microbes are classified under the *Bacillota* phylum, research has indicated that some species from *Actinomycetota*, *Pseudomonadota* and *Fusobacteriota* phyla can also generate butyrate [47,48].

Beyond their metabolic roles, SCFAs are also important in regulating anti-inflammatory pathways and modulating neuroinflammation [49,50,51].

The SCFAs can affect the secretion of hormones from enteroendocrine cells, such as cholecystokinin, peptide YY, and glucagonoid peptide [52]. Several studies have demonstrated the existence of direct synaptic communication between enteroendocrine cells and afferent neurons [53].

Specifically, projections (neuropods) of the entero-endocrine cells form “neuroepithelial cell units” with intestinal neuronal networks and directly connect the intestinal lumen to the nervous system, thus facilitating fast feedback to the enteroendocrine cells [54].

SCFAs can also cross the BBB and enter the brain, subsequently affecting neurotransmission by modulating the levels of various neurotransmitters as well as neurotrophic factors [55].

The effect of metabolites on BBB function is demonstrated by studies in sterile experimental animals that show increased BBB permeability. This increased permeability is mainly due to the disorganization of the occlusive junction proteins and the reduced expression of occludin and claudin 5. Recolonization of sterile experimental animals with SCFA-producing bacterial genera can re-enhance endothelial integrity [56,57].

Similarly, propionate can reverse BBB permeability after exposure to lipopolysaccharide (LPS), protecting the BBB from oxidative stress [58].

Butyrate also appears to restore BBB permeability and exerted a neuroprotective effect in an experimental brain injury model, increasing the expression of occludin and zonula occludens 1 [59].

Specifically, surplus butyrate that is not utilized by the colon or liver can be transported to the CNS, where it exhibits positive neurological impacts. For example, studies have demonstrated that oral administration of butyrate can alleviate depression and anxiety in mice, particularly in cases induced by chemotherapeutic agents, correlating with neuroprotective and anti-inflammatory effects [60].

Additionally, butyrate has been shown to enhance remyelination and mitigate neuroinflammation in the cerebral cortex caused by a high-fat diet in animal models [61]. High-fat diets are known to induce oxidative stress and hypothalamic inflammation. Furthermore, lower intestinal concentrations of SCFAs, including butyrate, are frequently observed in disease models associated with intestinal dysbiosis, which may play a role in the development and progression of neurological disorders and neuroinflammation [62,63].

Nonetheless, butyrate supplementation often fails to restore normal intestinal concentrations despite improvement in dysbiosis-related conditions. Whereas the specific mechanism by which butyrate influences neural physiology and behavior remains unclear, butyrate works as a dual communicator between the GBA by triggering the anti-inflammatory responses [64]. Butyrate helps maintain the integrity of the intestinal epithelial barrier. When the gut barrier is compromised, microbial components such as lipopolysaccharide (LPS) from Gram-negative bacteria enter systemic circulation. LPS can translocate into the brain, activating microglia and astrocytes, which promotes cytokine production and neuroinflammation. High levels of circulating LPS have been shown to activate microglia across various brain regions, resulting in synaptic loss and neuronal damage. Chronic pro-inflammatory stimuli from microglia can transform resting astrocytes into neurotoxic reactive cells, contributing to the degeneration of neurons and oligodendrocytes [65,66].

The absence of the GM, or even its less diverse composition, has a negative impact on microglial cells, rendering them incapable of inducing mature immune responses [67].

SCFAs can reduce the excessive activation of microglial cells and the production of pro-inflammatory cytokines. In addition, butyrate can induce an anti-inflammatory phenotype in microglial cells through morphological changes, elongating their protrusions and restoring their characteristic shape under normal conditions [68,69]. Interestingly, oral administration of propionate, acetate, and butyrate can repair defective microglia [67].

Bacteria metabolites may also have a regulatory role in astrocytes through the aryl-hydrocarbon receptor (AhR). Tryptophan is metabolized by the GM to indoles, which are AhR agonists, activating AhR signaling and reducing neuroinflammation in experimental models of autoimmune encephalomyelitis [70]. Studies also highlight the connection between the microbiota and myelination, reporting results of altered expression of myelination genes and levels of myelin proteins in the prefrontal cortex of sterile in vivo animals [71].

Finally, the GBA is interconnected with other microbiotas’ axes of the gut, such as the lungs, liver, heart, blood, urinary system, and others, and this can influence the function and neural structure of the CNS [23] (Figure 3)

## 6. The Role of Bioactive Lipids in Neuroprotection and Neuroplasticity Following Traumatic Brain Injury

Lipids are best known for their crucial roles as fundamental components of cellular membranes and as alternative energy sources. However, a specific group of lipids is gaining recognition for their important functions in regulating cell growth, adhesion, migration, signaling, and apoptosis [72]. These are referred to as bioactive lipids and are classified into four main families based on their biochemical activities: endocannabinoids, lysoglycerophospholipids/sphingolipids, classical eicosanoids, and specialized pro-resolving mediators (SPMs) [73,74].

Specifically, eicosanoids derived from AA, which is a type of omega-6 fatty acid (FA), are known to be pro-inflammatory bioactive lipids. They play a significant role in modulating both the intensity and duration of the inflammatory response [74]. Conversely, the bioactive lipids with inflammatory-resolving functions by inflammation downregulation represent a specific family called SPMs, which include resolvins, protectins, and lipoxin. Derived from the metabolism of FAs precursors, SPMs reduce neurological injury by serving as a molecular “stop signal” to neuroinflammation [75]. Collectively, these bioactive lipids are vital in managing both pro-inflammatory and anti-inflammatory responses, with SPMs playing a particularly significant role in minimizing neuroinflammation [73,74].

Preclinical studies indicate that these FAs play a crucial role in promoting neuroprotection and enhancing neuroplasticity. Omega-3 PUFAs are effective in inhibiting the transformation of microglia into a pro-inflammatory phenotype and in activating exosomal NGF/TrkA signaling pathways. Additionally, the same research demonstrated the suppression of apoptotic pathways, which resulted in a reduction in neuronal death [76,77].

Animal studies showed that omega-3 PUFA supplementation reduced neuroinflammation and improved cognitive outcomes when administered before and after TBI. Specifically, pre-injury treatments with omega-3 PUFAs resulted in reduced oxidative stress and improved spatial learning, while post-injury supplementation led to decreased markers of brain injury [78,79,80,81,82,83].

Moreover, omega-3 fatty acids contribute to maintaining the integrity of the blood–brain barrier, thereby reducing brain edema and preventing further neuronal damage [84].

In terms of neuroplasticity, omega-3 PUFAs contribute to clearance of waste products from the brain, which is essential for preserving neural health and function by supporting the glymphatic system. SPMs showed benefits in reducing inflammation and promoting recovery when administered intrathecally and intraperitoneally [81,83,85,86,87,88,89].

Case reports in human TBI patients suggest that high doses of omega-3 FAs may lead to positive neurological outcomes post-injury. However, no controlled randomized clinical trials have been conducted so far to evaluate the efficacy of PUFAs or SPMs in TBI patients. Although enteral feeding formulations containing omega-3 FAs are used in ICUs, their application in broader clinical practice remains limited, and future clinical trials are planned to further investigate this low-risk intervention [90,91,92].

The administration of lipids and lipidome characterization is essential for understanding metabolic disruptions after TBI. The lipidome reflects the diverse lipids in biological systems, offering insights into systemic metabolic changes within the first 24 h post-injury. Recent studies have highlighted the importance of specific choline phospholipids, such as lysophosphatidylcholines and sphingomyelins, as a marker of TBI severity. These findings suggest that metabolic changes may aim to restore lipid homeostasis in the brain, indicating that lipid administration could have therapeutic benefits. Investigating lipid-based interventions may further our understanding of their effects on systemic and cerebral health after TBI [93,94,95] (Table 1).

Several human studies have shown that a diet rich in saturated and trans fats increases the dysbiosis of the GM with a mainly anaerobic bacterial population together with the number of *Bacteroides* [96].

Additionally, modulation of the GM has been found to mitigate inflammation caused by metabolic endotoxemia in mice on a high-fat diet, suggesting potential benefits for the inflammatory processes associated with TBI [97].

## 7. The Role of Amino Acids in Neuroinflammation and Neuroprotection: Implications for Traumatic Brain Injury Recovery

Some amino acids exemplify the complex interplay between metabolism and inflammation in the nervous system. One amino acid that plays a central role in neuroinflammation is glutamate. Glutamate is the primary excitatory neurotransmitter in the brain and is involved in numerous processes, including synaptic plasticity and cognitive functions. However, excessive glutamate release and subsequent accumulation can lead to excitotoxicity, promoting neuroinflammation and contributing to neurodegenerative diseases [11].

Another amino acid of interest is arginine, which can be metabolized into nitric oxide (NO) by nitric oxide synthase in inflammatory processes. NO plays a role in various neuroinflammatory pathways, influencing neuronal signaling and immune response.

Additionally, tryptophan is significant, as it is a precursor to serotonin and can influence neuroinflammatory responses through its metabolites, including kynurenine, which has been implicated in neuroinflammation and neurodegenerative diseases.

Nutrition formulas containing amino acids such as glutamine (GLN) or arginine have been proposed to aid in the synthesis of biologically active peptides and to promote neuroprotection from secondary brain insults [98].

Glutamine is a non-essential amino acid primarily released from skeletal muscle, playing a critical role in protein synthesis and immune function, particularly in lymphocyte proliferation, macrophage activity, and neutrophil response, as an element of ATP biosynthesis and glycogenesis. Normally abundant in muscle tissue, GLN levels decline due to hypercatabolism from trauma or sepsis, alongside increased demands from various organs. This decline makes GLN “conditionally essential”, potentially impairing immune function and worsening clinical outcomes [99].

Studies have shown that GLN-enriched enteral diets reduce infections and length of stay in patients with moderate-to-severe TBI. However, when plasma glutamine levels are normal, excessive glutamine intake can lead to unfavorable outcomes. Given the dose dependence, ESPEN guidelines recommend not exceeding an enteral GLM supplement of 0.2–0.3 g/kg/day in critically ill trauma patients [100,101,102,103,104]. Future large-scale, multicenter, prospective RCTs that adopt standardized protocols are needed to strengthen the available literature.

Arginine is an amino acid that serves as a precursor of several active compounds, such as NO and creatine [105,106].

While arginine’s benefits are more related to its role in NO production and immune modulation, its direct protective effects in TBI are less established compared to creatine.

Creatine is known for its role in adenosine triphosphate (ATP) synthesis, which is crucial for energy metabolism in cells. Additionally, creatine has been shown to have neuroprotective properties. It may help reduce oxidative stress and apoptosis in neurons, which can be beneficial in a TBI scenario where neuronal injury occurs [107,108].

While primarily stored in skeletal muscle, creatine also supports the brain, which can synthesize creatine independently of dietary sources or other organs. The enzyme creatine kinase highlights its role in supplying energy to the central nervous system. Brain creatine deficiency is associated with major mental developmental disorders, and supplementation may alleviate these problems. Creatine aids recovery as a neuroprotective agent from the chronic manifestations that lead to oxidative stress and cognitive function post-brain injury [109,110,111].

TBI often leads to altered ATP demand due to decreased blood flow and hypoxia, reducing brain creatine levels. Research indicates that creatine supplementation could help combat these negative energy changes, potentially aiding recovery even years after the injury [69].

A study involving retired US NFL players showed a correlation between repetitive head impacts and decreased brain creatine, suggesting long-term disruptions in brain energy metabolism [112].

Experimental models and animal studies further support the hypothesis that creatine can improve cognitive function during episodes of oxygen deprivation and reduce brain damage from TBI. While humans typically experience only a modest increase in brain creatine from supplementation compared to animals, initial studies in humans show promise, indicating improvements in cognition, behavior, and physical symptoms associated with TBI in children [113,114].

Despite the lack of extensive clinical trials, creatine supplementation could be beneficial in medium TBI as a potential neuroprotective agent.

After TBI, the levels of branched-chain amino acids (BCAAs) such as valine, isoleucine, and leucine often decrease. This reduction may be attributed to several factors related to the increased metabolic demands of the brain following trauma, which can lead to an altered balance of amino acids in circulation. The brain is particularly sensitive to changes in amino acid availability, as these compounds are vital for neurotransmitter synthesis and overall neuronal health. BCAAs not only contribute to the maintenance of muscle mass and energy supply but also play roles in modulating inflammation, oxidative stress and neuroprotective properties, which are critical factors in the recovery process after a brain injury [115,116].

Diminished levels of BCAAs can also impair the synthesis of glutamate and gamma-aminobutyric acid (GABA), two key neurotransmitters that influence excitability in the brain. This imbalance may lead to increased neuronal excitability, contributing to issues such as seizures, mood disorders, and cognitive difficulties, all of which are prevalent in individuals recovering from TBI. Supplementation with BCAAs may enhance metabolic support for brain recovery, potentially mitigating some of the negative consequences of severe and mild TBI [115,117,118].

Further investigations are needed to examine the impact of BCAA supplementation on patients with severe or mild TBI. It is also important to investigate how supplementation effects may vary based on patient characteristics, including age, gender, cognitive abilities, and emotional and behavioral conditions.

Tryptophan, an essential amino acid, plays a significant role in modulating the gut–brain axis through its metabolites, which are implicated in neuroinflammation and neurodegeneration [119]. During bacterial fermentation, tryptophan is metabolized into various indole-containing compounds, many of which act as aryl hydrocarbon receptor ligands [78].

Furthermore, tryptophan-derived metabolites serve as ligands for the aryl hydrocarbon receptor (AhR), a transcription factor found in diverse cell types, including dendritic cells, microglia, and astrocytes. The presence of AhR ligands in cerebrospinal fluid indicates that these microbiota-derived metabolites may exert direct effects on CNS resident cells, particularly microglia and astrocytes, which express AhR. Recent findings suggest that AhR signaling mediated by these ligands can restrain the production of inflammatory cytokines, such as interleukin 6 (IL-6), tumor necrosis factor alpha (TNF-α), and inducible nitric oxide synthase (iNOS), in astrocytes, highlighting the potential therapeutic implications of tryptophan metabolites in neuroinflammatory conditions [70,120].

While some studies indicate potential benefits, the evidence is not yet conclusive [121,122]. More clinical trials are needed to fully understand the efficacy and safety of tryptophan supplementation in TBI patients (Table 2).

## 8. The Role of Carbohydrates in Traumatic Brain Injury

Carbohydrates are one of the three primary macromolecules, alongside lipids and proteins.

If more carbohydrates are consumed than the body needs or can store as glycogen, the excess will be stored in adipose tissue. This includes simple carbohydrates or sugars, such as monosaccharides like glucose and fructose, as well as disaccharides like sucrose and lactose [123].

Complex carbohydrates (oligosaccharides and polysaccharides) are based on three simple ingredients: fiber, starch, and sugar. As such, their nutritional value depends on the proportion of each in the ingredients.

The role of carbohydrates in the body is quite important. They can be an immediate source of energy through the conversion into glucose and are part of DNA thanks to ribose. The five-carbon monosaccharide ribose is an important component of coenzymes (e.g., ATP, FAD, and NAD) and the backbone of the genetic molecule RNA [124].

As a result, they are an ideal energy source for the brain. The complex carbohydrates found in foods rich in fiber and nutrients help stabilize blood sugar levels. Some examples of nutritious carbohydrates include fruits, vegetables, whole grains, nuts, and legumes. These carbohydrates are metabolized slowly, allowing for the absorption of essential nutrients. Given that the brain is a vital organ that requires a certain amount of carbohydrates to operate effectively, it is reasonable that they play a significant role in a balanced diet [125].

Complex carbohydrates, particularly those rich in dietary fiber, are prebiotics that can significantly influence GM composition, promoting the growth of beneficial bacteria such as *Lactobacillus* and *Bifidobacterium*. These beneficial bacteria ferment complex carbohydrates into SCFAs, the properties of which have previously been described [43].

On the other hand, simple carbohydrates are absorbed faster by the body, which leads to a rapid energy supply; however, this effect seems to have negative consequences for the host. The initial increase in energy quickly results in a sharp decrease, which affects blood sugar levels and can lead to dysmetabolism due to long-term dysbiosis of the GM. This dysbiosis may result in local and then peripheral endotoxemia, contributing to metabolic syndrome (such as diabetes and atherosclerosis) and ultimately impacting all GM axes, particularly the gut–brain axis [126,127,128]. The brain requires 110–145 g of glucose daily, utilizing about 20% of the body’s energy, primarily for neuron activation, including during sleep. After a TBI, glucose metabolism is disrupted, with insulin playing a significant role, particularly in the motor cortex. Approximately 87% of ICU patients with TBI experience hyperglycemia. While the mechanisms behind post-TBI elevated blood glucose are not fully understood, there is a noted connection to tissue lactic acidosis, which is associated with increased mortality. The injured brain demands more energy, relying on oxidative metabolism of glucose and oxygen, and a lack of these can lead to neurological decline or death [129]. Common effects of neurotrauma on glucose metabolism include hyperglycolysis, mitochondrial dysfunction, and varying cerebral metabolic demands. Excessive glucose or carbohydrate administration beyond the body’s oxidation capacity (5 mg/kg/min or 0.3 g/kg/h) can worsen hyperglycemia, reflecting the severity of the stress response and primary injury, with ICU management potentially aggravating the situation [130].

## 9. Functional Foods: Symbiotics (Probiotics/Psychobiotics and Prebiotics) and Metabiotics (Postbiotics and Parabiotics)

Probiotics are live microorganisms, mainly bacteria and yeasts, that provide health benefits to the host when consumed in adequate amounts. Known sources of probiotics include yogurt, kefir, and certain other fermented foods. The most used organisms in these sources include *Streptococcus* spp. (such as Streptococcus thermophilus), *Enterococcus* spp., *Bifidobacterium* spp., and species that belong to the *Lactobacillaceae* family (such as *Lactobacillus reuteri*, *Lactiplantibacillus plantarum* and other) [131].

Probiotics modulate the GM, impacting various pathways that connect the gut to the brain and other systems (Figure 4). This modulation enhances gut barrier integrity, preventing harmful substances from crossing into the bloodstream. Through the endocrine system, probiotics influence the release of hormones like ACTH and cortisol, which regulate stress responses. In the immune system, they reduce inflammation by modulating cytokine production, crucial for controlling systemic and neuroinflammation. Probiotics also affect neurochemical pathways, boosting the production of neurotransmitters such as serotonin, dopamine, and GABA [132]. Several studies have shown that this boosts gut health and overall anti-inflammatory and immune responses [133].

Various formulations of probiotic products are available, ranging from bacterium species to yeasts, such as *Saccharomyces cerevisiae* and *Saccharomyces Boulardii* and even *Aspergillus oryzae*, which is a filamentous fungus, which may be available as capsules, powders, pastes, tablets, or sprays depending on need. The use of probiotics is a more natural approach and has fewer side effects compared to pharmacotherapeutic drugs [134].

Some probiotics are psychobiotics because there are live microorganisms with mental or organic health benefits through the modulation of the GM [135].

Thus, they can also be called neurobiotics. Those beneficial microorganisms can modulate the GBA, influencing the stress response, neurotransmitter production, and immune function. The most promising strains include several bacterial species from the *Lactobacillaceae* family, such as *Lactobacillus rhamnosus*, which targets stress through GABA receptors. Thus, the GM interacts with the GBA by modulating sensory afferents and the immune system [22].

*Lactobacillus reuteri* modulates intestinal motility and pain perception by inhibiting the opening of calcium-dependent potassium channels. In addition, the GM can influence ENS activity by producing molecules that act as local neurotransmitters, such as GABA, serotonin, melatonin, histamine, and acetylcholine. It can also produce a biologically active form of catecholamines in the gut lumen via enterochromaffin cells [136,137].

*Bifidobacterium longum* moderates the amount of cortisol and mental/physical stress. *Lactiplantibacillus plantarum* (old name *Lactobacillus plantarum*) affects dopamine and serotonin levels to enhance mood [138].

The effects of psychobiotics are related to restoring tight junction integrity and protecting the intestinal barrier. It has been observed that pretreating animals with a psychobiotic bacterium-combination formulation of *Lactobacillus helveticus* R0052 and *Bifidobacterium longum* R0175 restored the integrity of the tight junction barrier and moderated the activities of the HPA axis and the ANS, as assessed by plasma cortisol and catecholamine measurements. Probiotics prevented changes in hippocampal neurogenesis and expression of hypothalamic genes involved in synaptic plasticity [139].

Thus, these probiotics, which regulate beneficial gut bacteria and enhance gut barrier integrity, have shown protective effects post-TBI [7]. *Lactobacillus acidophilus* supplementation improved intestinal barrier function and nutrient absorption after TBI, and restored gut motility in animal studies [140,141,142,143]. The administration of *Limosilactobacillus reuteri* in military veterans with mild TBI showed significant reductions in inflammatory markers [143]. A randomized clinical trial showed that daily prophylactic administration with probiotics reduced infection rate in TBI patients [144]. Finally, psychobiotics have beneficial effects on immune modulation in relation to any kind of neuroinflammation via the GBA. A probiotic compound with *L. acidophilus*, *Bifidobacterium bifidum*, *L. reuteri*, and *Limosilactobacillus fermentum* (old name *Lactobacillus fermentum*) decreased the expression of the pro-inflammatory genes IL-1, IL-8, and TNF-α and increased the expression of TGF-β and PPAR-γ, which is associated with anti-inflammatory processes. It has also been noted in animal models that gliosis can be prevented by the strain of *L. plantarum* PS128 because it has a beneficial effect against neuroinflammation and improves cognitive function.

Several studies have shown that fecal microbiota transplantation and probiotics can improve certain neurological damage and dysfunction in animal models of stroke and spinal cord injury. In addition, probiotics have been shown to reduce both the incidence of infections and the length of stay in intensive care in patients with brain trauma [145,146,147].

Prebiotics are indigestible compounds consisting of fructooligosaccharides (FOS) and galactooligosaccharides (GOS) that stimulate bacterial growth. *Bifidobacteria* spp. and species from the *Lactobacillaceae* family have been found to increase with consumption of FOS and GOS, and consumption of prebiotics has been suggested as a possible treatment for intestinal dysbiosis [148].

In addition to modulating the GM, anti-inflammatory and antioxidant effects of prebiotics have been suggested. The combination of probiotics and prebiotics are symbiotics, and they are usually applied due to their synergistic effect in food [149].

Therefore, postbiotics are produced by living microorganisms with the advantage of stability and transportability, so they have a better shelf life and easy packaging. Therefore, they can be used as functional elements in products such as dairy products, vegetables, bread, meat, fish products, food ingredients etc. Their effect is positive in immunomodulatory, antitumor, and antimicrobial respects. On the other hand, paraprobiotics, also known as “non-viable probiotics”, refer to inactivated (non-viable) microbial cells that provide health benefits when administered in appropriate doses [150].

Metabiotics is the combined use of prebiotic and parabiotic ingredients for synergistic action in food. Metabiotics have biological activity and provide benefits to the host. The advantage of metabiotics lies in their safety profile and longer shelf life over probiotics, while providing health benefits comparable to those of probiotics [151,152].

In that context, the new evolution of functional foods is represented by metabiotic products, which refer to the by-products or metabolic substances produced by probiotic microorganisms during fermentation. These by-products include SCFAs, enzymes, peptides, organic acids, and polysaccharides [153,154] (Table 3).

The administration of *C. butyricum*, a butyrate-producing bacterium, showed neuroprotective effects by improving neurological outcomes and preserving brain integrity in mice [155]. Overall, these findings underscore the potential therapeutic role of probiotics in TBI management, warranting further research into their effects on brain inflammation and TBI-related behaviors such as cognitive impairments, emotional dysregulation, sleep disturbances, and impulsivity.

Finally, bacteria tryptophan metabolites (indole and indole derivatives) modulate microglial and astrocytic functions, LPS, and peptidoglycan are microbe-associated molecular patterns. Nevertheless, more studies are needed to validate the beneficial effects of symbiotics and metabiotics either alone or in combination with probiotics [153,156].

## 10. Conclusions

The interplay among the microbiota, macronutrients, and neuroinflammation offers a promising framework for advancing our understanding and treatment of TBI. The biochemical pathways activated following TBI remain complex, influenced by an array of metabolic, inflammatory, and microbial factors. Dietary macronutrients show potential in attenuating the neuroinflammatory cascade and promoting gut microbiome health, thus enhancing the brain–gut axis’s functional integrity. Emerging evidence suggests that specific dietary interventions might mitigate the severity of secondary brain injury while fostering neurological recovery. However, significant gaps remain in clinical research, necessitating well-designed trials to establish efficacy and safety. As our comprehension of gut–brain interactions evolves, future therapeutic strategies should increasingly consider tailored nutritional approaches as integral components of TBI management. Integrating nutritional optimization into clinical practice can enhance TBI patients’ recovery trajectories, ultimately improving both neurological outcomes and quality of life.

## Figures and Tables

**Figure 1 nutrients-16-04359-f001:**
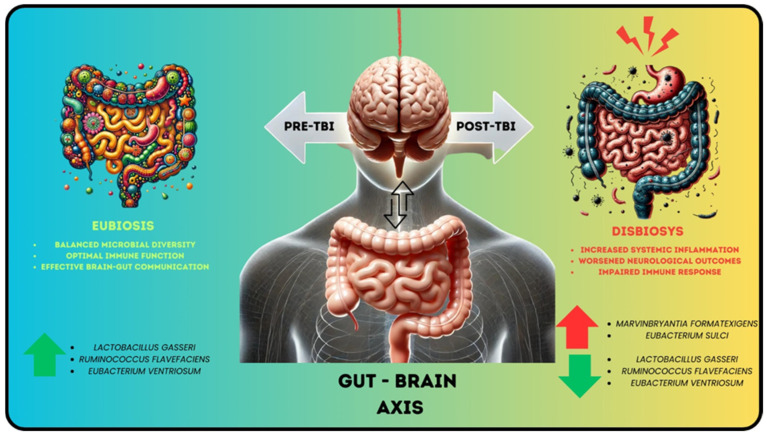
The gut microbiota undergoes significant changes following traumatic brain injury (TBI), characterized by the depletion of beneficial species such as *Lactobacillus gasseri* and *Eubacterium ventriosum* and an increase in potentially harmful species like *Marvinbryantia formatexigens* and *Eubacterium sulci*. This dysbiosis may exacerbate systemic inflammation and negatively impact neurological recovery through the gut–brain axis.

**Figure 2 nutrients-16-04359-f002:**
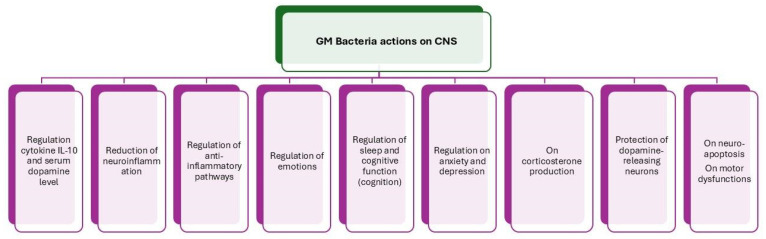
The main biochemical actions by the GM-friendly bacteria on the CNS.

**Figure 3 nutrients-16-04359-f003:**
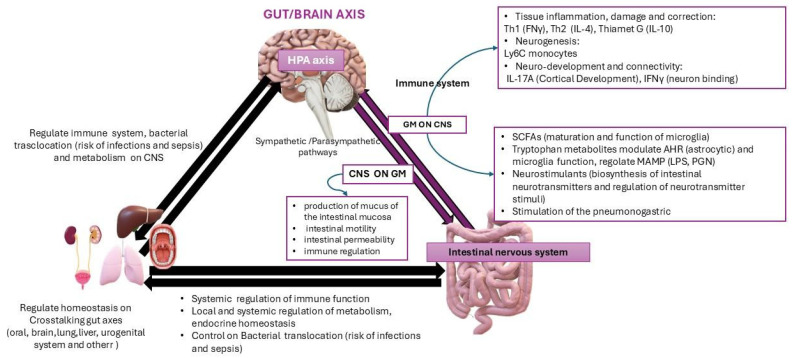
The figure demonstrates the influence of the CNS on the gut microbiota and vice versa. It also illustrates the hypothesis of the interconnection between the gut–brain axis and other axes of the microbiota and their influence on homeostasis for the health of the host.

**Figure 4 nutrients-16-04359-f004:**
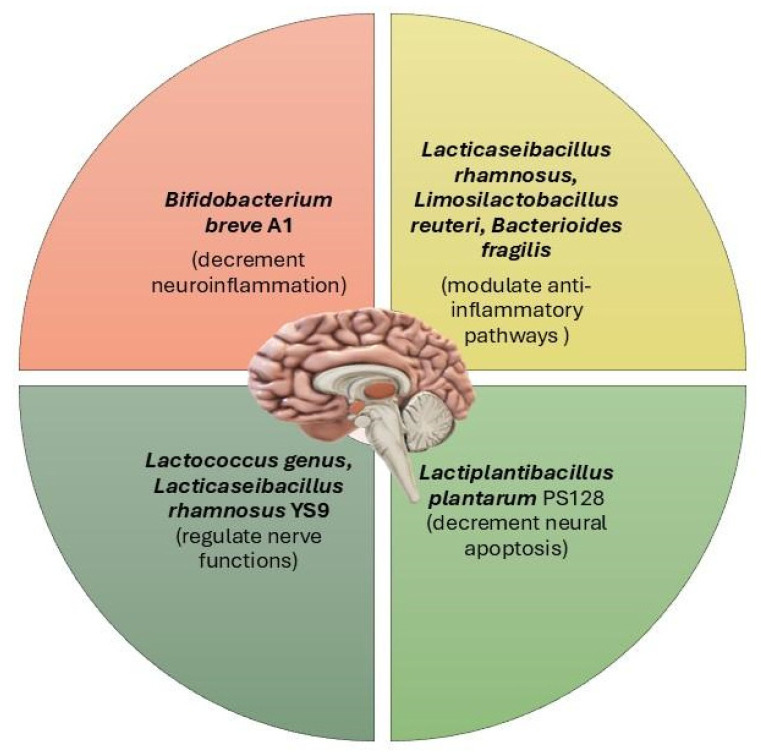
Some probiotic bacteria that influence the function and structure of neurons.

**Table 1 nutrients-16-04359-t001:** Role, health benefits, and therapeutic potential of lipids and short-chain fatty acids in neuroprotection and inflammation management.

Role of Lipids	
**Cell Membrane Structure & Energy Source**	Lipids are crucial for cellular membrane integrity and energy storage
**Bioactive Lipids**	Emerging focus on bioactive lipids, categorized into four families: Endocannabinoids Lysoglycerophospholipids/Sphingolipids Classical Eicosanoids (Pro-inflammatory) Specialized Pro-Resolving Mediators (SPMs) (Anti-inflammatory)
**Benefits of Omega-3 Fatty Acids (PUFAs)**	**Neuroprotection & Neuroplasticity** - Inhibition of pro-inflammatory microglial activation. - Enhancement of neurotrophin signaling (exosomal NGF/TrkA). - Suppression of apoptotic pathways, thereby reducing neuronal death **Cognitive Improvement in TBI** *Animal studies indicate that omega-3 supplementation before/after traumatic brain injury (TBI):* - Reduces oxidative stress and neuroinflammation. - Improves cognitive functions and spatial learning. - Maintains blood-brain barrier integrity, reducing edema
**Benefits of Specialized Pro-Resolving Mediators (SPMs)**	Inflammation Resolution: SPMs play a critical role in transitioning from pro-inflammatory to anti-inflammatory states, potentially improving outcomes after neuroinflammatory damage
**Benefits of Short-Chain Fatty Acids (SCFAs)** - Acetate - Propionate - Butyrate	Neuroinflammation Modulation:SCFAs have significant roles in regulating central nervous system (CNS) inflammation and overall brain function: - Butyrate alleviates depression and anxiety, enhances remyelination, and acts against neuroinflammation.

**Table 2 nutrients-16-04359-t002:** Schematic summarizing the roles of the highlighted amino acids in metabolism, neuroinflammation, and potential therapeutic implications, particularly in the context of traumatic brain injury.

Amino Acid Role on Nervous System			
Amino Acids	Role	Impact on Health	Potential Therapy
**Glutamate**	Primary excitatory neurotransmitter; critical for synaptic plasticity and cognitive function.	Excessive glutamate can lead to excitotoxicity, triggering neuroinflammation and contributing to neurodegenerative diseases.	Glutamate-related interventions may aid in neuroprotection against brain insults.
**Arginine**	Precursor to nitric oxide (NO), involved in inflammatory pathways.	Modulates immune responses and neuronal signaling relevant to neuroinflammation.	While its effects in TBI need further study, arginine’s role in NO production poses potential benefits.
**Tryptophan**	Precursor to serotonin and involved in gut-brain signaling through metabolites influencing neuroinflammation.	Metabolites (like kynurenine) can be linked to neuroinflammation and neurodegenerative diseases	Potential benefits warrant further investigation in clinical settings.
**Glutamine**	Supports protein synthesis and immune function; may become “conditionally essential” during hypercatabolic states.	May reduce infections and length of hospital stay in TBI patients	Recommended supplementation should not exceed 0.2–0.3 g/kg/d to avoid adverse effects.
**Creatine**	Involved in ATP synthesis and energy metabolism; supports cellular energy supply.	Demonstrated neuroprotective properties; may reduce oxidative stress and apoptosis in neurons, beneficial in post-TBI recovery.	Creatine supplementation has shown promise for improving cognitive function and recovery following TBI.
**Branched-Chain Amino Acids (BCAAs)**	Includes valine, isoleucine, and leucine; crucial for maintaining muscle mass and energy, and modulating inflammation.	Reduced BCAA levels post-TBI could impair neurotransmitter synthesis, potentially leading to cognitive and mood disorders.	Supplementation may support brain recovery and improve overall metabolic balance.

**Table 3 nutrients-16-04359-t003:** Some metabiotic bacteria strains that are used as components related with their action.

Metabiotic Component Bacteria Strains				
Neurotransmitters				Metabolome
Dopamine	GABA	Serotonin	Histamine	Metabolites/SCFAs *
*Bifidobacterium bifidum* *Bifidobacterium longum* *Lacticaseibacillus rhamnosus* *Lacticaseibacillus rhamnosus* GG *Lactiplantibacillus plantarum* LP28 *Lactiplantibacillus plantarum* PS128 *Limosilactobacillus reuteri* ATG-F4 *Lactococcus lactis* subsp. *Lactis*	*Lacticaseibacillus rhamnosus* JB-1 *Lacticaseibacillus rhamnosus* YS9	*Enterococcus* *Escherichia* *Lactococcus* *Streptococcus* *Lactobacillaceae* family	*Enterococcus* *Escherichia* *Lactococcus* *Streptococcus* *Lactobacillaceae* family	*Bacteroides fragilis* NCTC 9343 *Bifidobacterium breve* A1 * *Lacticaseibacillus rhamnosus* * *Limosilactobacillus reuteri* * *Bacterioides fragilis*

## Data Availability

Data are contained within the article.

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
