# Peer review of "The Role of Macronutrients and Gut Microbiota in Neuroinflammation Post-Traumatic Brain Injury: A Narrative Review"

_nutrients, 2024, doi:10.3390/nu16244359_

Round 1
Reviewer 1 Report
Comments and Suggestions for Authors
Cotoia et al review the role of macronutrients and gut microbiota in neuroinflammation and post-traumatic brain Injury. The topic was well reviewed although the paper appears too lengthily with a long general introduction to the subject. The illustrations are adequate. Below are some specific comments.
l. 21, replace and by as well as since probiotic is not a macronutrient
l. 49 facilitates
l. 63-65 revise the sentence
l.101 what do you want to say, proteins are released from cytoplasm and nucleus or taken up into these structures?
l.110 this is a duplicate of PUFA abbreviation and with a larger font size
l. 154 rather by enteroendocrine hormones than cells
l. 156 do you mean efferent?
l.158-159 mentioning the ANS here is misplaced as long as you already speak about the n.vagus
l. 184 this is the 1st time you mention ENS
l. 215-219 check the formatting, and use CRF or CRH in above sentences
L. 223 Enterobacteriaceae fam, such as Enterococcus spp., this is wrong taxonomy, please correct
L. 230, 231 I do not think that ‘expression’ is a correct term for these sentences
L236-237, sentence is not clear
L. 239 should be neurotransmission, also writing microbiotas’ is rather unusual for this term
L. 248. Check the sentence
L.257 Bacterial
L. 272, refs 52, 53 are referring to other reviews, here and in other places where you give specific information, the review should provide references to the original articles
L. 368 ‘specialized SPM‘ is a duplicate of S in this abbreviation, from the other hand, it is not so clear about what mediators do you speak about here
L.412-413 Check the sentence
L.487-489. Check the sentence
L. 574 What are the behaviors related to brain inflammation?
Reviewer 2 Report
Comments and Suggestions for Authors
In the present work, Cotoia et al. describe the role of macronutrients and gut microbiota in ameliorating traumatic brain injury (TBI). The article is overall well-written and interesting. The quality and number of figures and tables is well. However, i feel that deepening on some important aspects on this topic will provide an increased value to the present manuscript
1. Despite aminoacids and lipids are covered as important macronutrients, little is found about complex carbohydrates, although dietary fiber is mentioned in the section 5. However, i think that it will be necessary another section exploring in greater deep this important nutrient.
2. Despite is really interesting to talk about beneficial nutrients, it is also important to cover nutrients related to harmful effects on TBI such as simple carbohydrates or trans fatty acids due to their detrimental effects on gut microbiota. I thin it should be covered here as well
3. For a more practical perspective i feel that it should be also adviseable to include data related to specific dietary patterns, like mediterranean diet, rich in beneficial macro and micronutrients. Perhaps in the section of conclusions, some recommendations related to these alternatives will provide a great value for the readers, in aim to include potential interventions in the clinical management of these patients.
4. Please, the references should be provided in tables.
Minor:
GBA axis= Gut brain axis, not brain gut axis, please check
